# Structure and exfoliation mechanism of two-dimensional boron nanosheets

Jing-Yang Chung[1,2,7], Yanwen Yuan[1,2,7], Tara P. Mishra [1], Chithralekha Joseph[1], Pieremanuele Canepa [1], Pranay Ranjan[3], El Hadi S. Sadki [4], Silvija Gradečak [1,2] ✉ & Slaven Garaj [1,5,6] ✉

Exfoliation of two-dimensional (2D) nanosheets from three-dimensional (3D) non-layered, non-van der Waals crystals represents an emerging strategy for materials engineering that could significantly increase the library of 2D materials. Yet, the exfoliation mechanism in which nanosheets are derived from crystals that are not intrinsically layered remains unclear. Here, we show that planar defects in the starting 3D boron material promote the exfoliation of 2D boron sheets—by combining liquid-phase exfoliation, aberration-corrected scanning transmission electron microscopy, Raman spectroscopy, and density functional theory calculations. We demonstrate that 2D boron nanosheets consist of a planar arrangement of icosahedral sub-units cleaved along the {001} planes of β-rhombohedral boron. Correspondingly, intrinsic stacking faults in 3D boron form parallel layers of faulted planes in the same orientation as the exfoliated nanosheets, reducing the {001} cleavage energy. Planar defects represent a potential engineerable pathway for exfoliating 2D sheets from 3D boron and, more broadly, the other covalently bonded materials.

Two-dimensional (2D) materials—defined by the extreme aspect ratio of the thickness *vs.* lateral dimensions—are traditionally derived from a limited library of three-dimensional (3D) van der Waals (vdW) solids. The vdW materials consist of stacked layers with strong in-plane atomic bonds and weak inter-layer coupling (Fig. 1a), which could be separated through physical or chemical means to form 2D nanosheets[1-3]. This process inherently narrows the range of accessible 2D materials to a finite set of starting vdW materials. The recent observation of 2D nanosheets and nanoplatelets derived from non-layered non-vdW crystals[4-17], i.e., crystals with strong chemical bonding in all three directions, could significantly broaden the field of low-dimensional systems. In the exfoliation process[5], the starting non-vdW crystals with both high[6-8] and—surprisingly—low[11-17] bonding anisotropy (i.e., isotropic materials) can lead to the formation of 2D nanosheets and nanoplatelets. Yet, based on surface energy

considerations, it would be expected that isotropic crystals give rise to 3D nanoparticles, not 2D nanosheets—making the exact nature of the exfoliation mechanism the most pressing question in the nascent field[5]. Answering this question would enable the rational design of a new class of 2D materials from other, hitherto untapped isotropic materials.

Among the 2D structures exfoliated from non-layered materials, boron nanosheets[17-26] have drawn particular interest due to their potential for a range of applications, including energy storage[17], catalysts[18,19], drug delivery[20], photoelectronics[21,23], bioimaging[22-24], and sensors[26]. Boron is often referred to as a 'rule breaker' because of its electron deficiency with multi-center covalent bonds, resulting in a range of crystal phases with unusual stoichiometry and chemical bonding[27,28]. Whilst 3D boron forms allotropes comprised primarily of $B_{12}$ icosahedral units, 2D boron can have multiple phases[29,30], including

[1]Department of Materials Science and Engineering, National University of Singapore, Singapore, Singapore. [2]Applied Materials - NUS Advanced Materials Corporate Lab, National University of Singapore, Singapore, Singapore. [3]Department of Metallurgical and Materials Engineering, Indian Institute of Technology Jodhpur, Jodhpur, Rajasthan, India. [4]Department of Physics, College of Science, United Arab Emirates University, Al-Ain, UAE. [5]Department of Physics, Centre for Advanced 2D Materials, National University of Singapore, Singapore, Singapore. [6]Department of Biomedical Engineering, National University of Singapore, Singapore, Singapore. [7]These authors contributed equally: Jing-Yang Chung, Yanwen Yuan. ✉e-mail: gradecak@nus.edu.sg; slaven@nus.edu.sg

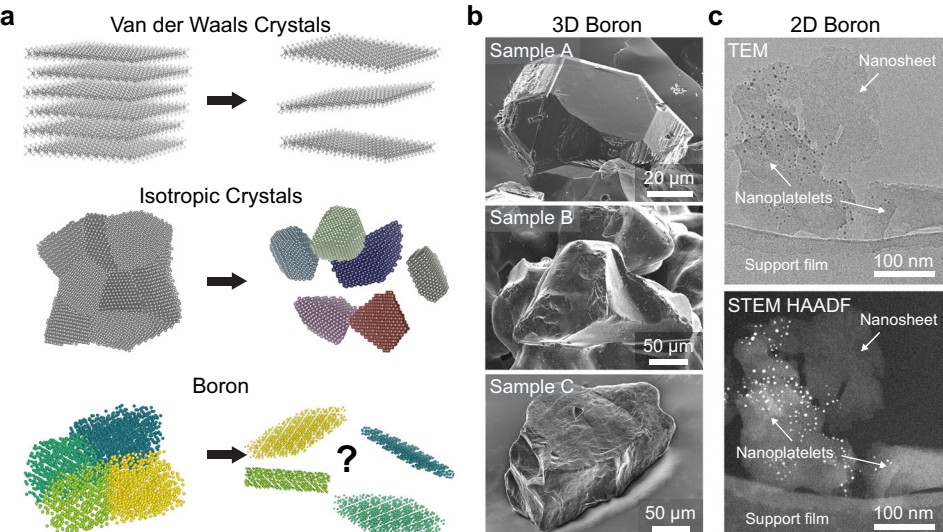

**Fig. 1 | Liquid phase exfoliation of boron. a** Schematic depiction of products through liquid-phase exfoliation from van der Waals (vdW) crystals, isotropic polycrystalline materials, and boron. **b** Scanning electron microscopy (SEM) images of the starting boron material: Sample A (Sigma-Aldrich), Sample B (Alfa-Aesar), and Sample C (Yamanaka Advanced Materials). **c** Converged beam transmission electron microscopy (TEM) image, and corresponding high-angle annular dark-field (HAADF) scanning transmission electron microscopy (STEM) image of 2D boron sheets suspended on a holey carbon TEM grid.

the synthetic graphene-like vacancy-decorated hexagonal monolayers, termed borophene[31]. Therefore, unlike its isotropic counterpart, 2D boron can uniquely display both metallic and semiconducting character[28], opening the possibility for an all-boron electronic device.

To date, both bottom-up (chemical and physical vapor deposition)[26,29–37] and top-down (ultrasonication and mechanical exfoliation)[17–25,38] syntheses have led to boron nanosheets with purportedly different crystal phases and thicknesses ranging from a monolayer to 10 nm. In the case of bottom-up approaches, the boron layer grown on metallic substrates has a monoatomic, hexagonal borophene structure[31,33–35], as revealed by scanning tunneling microscopy. On the other hand, many competing structural models of 2D-boron nanosheets synthesized through liquid-phase exfoliation (LPE) have been proposed, including β-rhombohedral icosahedral planes inherited from the 3D structure[17,19–23,25], or the hexagonal monolayer phases of β_{12} and χ_3[18,24]. LPE is a method for separating layers from a macroscopic material[1] using ultrasonic waves or vortexes to generate shear forces in a liquid medium and is not expected to alter the phase of the material. Considering that 3D boron is non-layered and has no vdW gap[28], it is surprising that LPE results in the formation of 2D sheets, rather than nanoparticles typical of isotropic polycrystalline materials (Fig. 1a).

Here, we provide a general framework for understanding the formation of 2D nanosheets, by showing that the crystal planes of the nanosheets correspond to the orientations of planar defects observed in the starting 3D boron phase. This was achieved by visualizing the phases, orientation, and internal structures of both the starting boron material and 2D nanosheets down to the atomic level using aberration-corrected scanning transmission electron microscopy (STEM). Through correlation with atomic models and multi-slice STEM simulations, we show that the starting 3D boron material and 2D nanosheets have the same icosahedral β-rhombohedral structure, where nanosheets consist of a few layers of the icosahedral planes following the direction of the planar defects. Our STEM results are corroborated by density functional theory (DFT) calculations, affirming the observed defect to possess the lowest cleavage energy among all β-rhombohedral cleavage planes—and together, they suggest that defects mediate the exfoliation mechanism. These findings provide the groundwork for future exfoliation of low-dimensional sheets from

other non-vdW materials, thus providing a new paradigm for engineering 2D materials.

We show that the number of layers (i.e., thickness) in a nanosheet and small variations in the viewing angle significantly affect the resulting STEM image, due to the overlapping structural icosahedral-based units. Careful experimental observations coupled with the STEM simulations are required to avoid misinterpreting the results and to precisely determine the 2D boron crystal structure.

## Results
### Exfoliation of boron nanosheets from diverse 3D crystals
To elucidate the exfoliation mechanism of boron 2D sheets in relation to materials-specific properties, we focus on a range of starting boron materials from diverse sources: Samples A, B, and C were obtained from Sigma-Aldrich, Alfa-Aesar, and Yamanaka Advanced Materials, respectively. These were selected as commercially available bulk boron materials, some of which were used in previous LPE studies[17,20,21,24,25], and we show they have various structural properties, including grain size and defect density. Examples of the particle sizes used for the liquid-phase exfoliation of Samples A, B, and C are shown through scanning electron microscope (SEM) images in Fig. 1b. Contrasting the distinctive faceted morphology of Sample A, the particles of Samples B and C display rougher surface morphology. From polarized light microscope images shown in Supplementary Information, Fig. 1a, b, the grain sizes of Samples A and B are observed to be within the same magnitude (mean diameter: 148.2 ± 78.9 μm and 105.3 ± 30.8 μm), while Sample C is roughly a magnitude smaller (11.0 ± 3.7 μm). Samples A and B exhibit straight lines indicative of planar defects cutting across the grains (Supplementary Fig. 2), with closer-spaced defects visible in Sample B. Sample C exhibits the highest defect density, which can be viewed more clearly through electron backscatter diffraction (EBSD) as well as strain-contrast transmission electron microscope (TEM) and STEM images (Supplementary Figs. 1a and 3, respectively). Liquid-phase exfoliation of the three boron samples was conducted using acetone as the sonication medium (20 g boron per 10 ml solvent) for 16 h, similar to previous reports that have demonstrated boron exfoliation[19,24]. Following the exfoliation, the sonicated products of Sample A contain mostly 3D crystallites, whereas those of Samples B and C comprise a combination

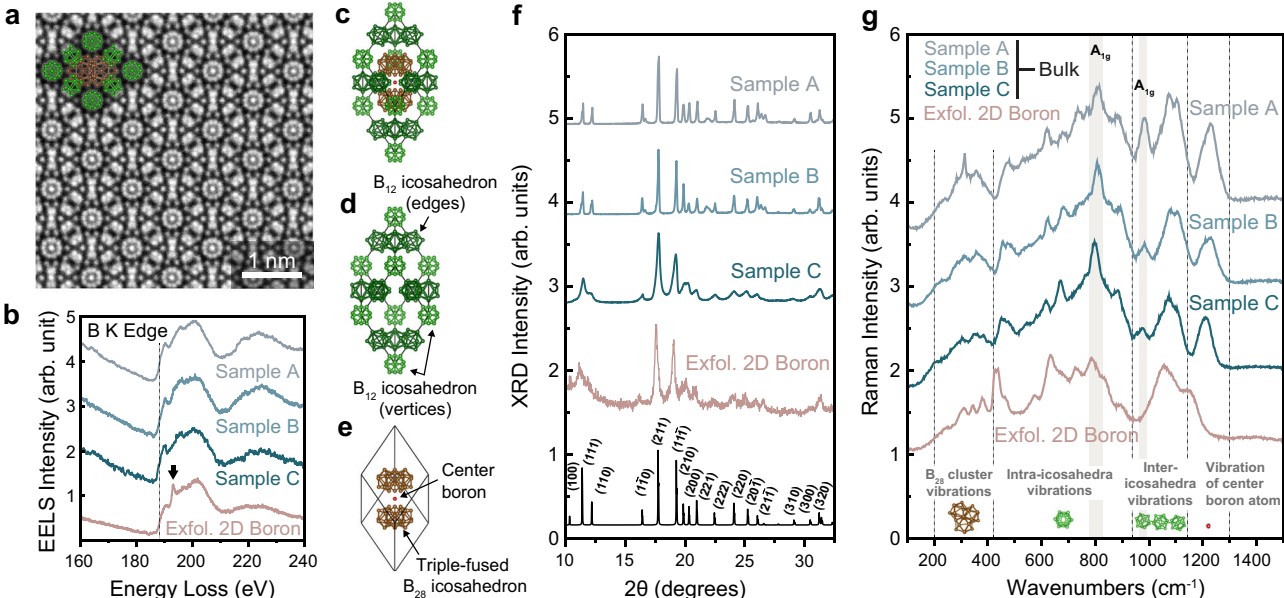

**Fig. 2 | Phase and elemental characterization of 3D boron and exfoliated 2D boron. a** Atomic-resolved center bright-field (C-BF) STEM image of 3D boron revealing the β-rhombohedral phase viewed along the [100] zone-axis. **b** Electron energy loss spectroscopy (EELS) spectra of 3D and exfoliated 2D boron, with the B K edge marked by the dashed line, and the B-O near-edge structure marked by the arrow. **c** Schematic unit cell of β-rhombohedral boron (projected in the [11$\bar{2}$] orientation), which can be represented separately in terms of (**d**) $B_{12}$ icosahedra located at the vertices (light green) and middle of each edge (dark green) of the rhombohedron, and (**e**) two $B_{28}$ triple-fused icosahedra (brown) + one boron atom (red) centrally located in the middle of the rhombohedron. **f** X-ray diffraction (XRD) patterns of 3D Samples A, B and C and exfoliated Sample C. The simulated pattern (in black) is shown indicating Miller indices of rhombohedral boron. The curves have been vertically shifted for clarity. **g** Raman spectra of 3D boron and exfoliated Sample C. Regions corresponding to the vibrations of the individual unit cell sub-units are segmented by the vertical dashed lines and displayed at the bottom. The shaded regions denote the $A_{1g}$ modes. The curves have been vertically shifted for clarity.

of smaller crystallites, wedged platelets, and 2D sheets (see Supplementary Fig. 4). Considering the larger grain sizes in Sample A, the lack of nanosheets in this case thus indicates a possible relation between the exfoliation efficiency and the internal structure of the starting material.

An example of a 2D boron sheet exfoliated from Sample C is shown in Fig. 1c. The low dimensionality of the nanosheets is evidenced by the lack of observable Kikuchi patterns when the nanosheets are imaged under converged beam TEM at the back-focal plane (Fig. 1c, top). The low dimensionality is further corroborated by the low contrast of the nanosheet with respect to the supporting carbon film (which is typically below 10 nm thick) when imaged using high-angle annular dark-field (HAADF) STEM (Fig. 1c, bottom). For further description of the TEM/STEM analysis and comparison against thicker platelets, see Supplementary Figs. 5 and 6. AFM imaging of 100 flakes likewise demonstrates their mean thickness to be 3.9 nm (Supplementary Fig. 7). This thickness-screening and cross-correlation between TEM and AFM approaches is essential as we find that solvent/surfactant residue can erroneously give the appearance of 'sheets'[5] (for example, see Supplementary Figs. 8 and 9). STEM HAADF imaging thus provides an efficient means to rapidly survey nanosheet dimensions, ensuring that phase characterization is not unintentionally performed on thicker nanoplatelets, rather than on 2D nanosheets.

**The crystal phase of 3D boron**
We next focus on the atomic structure of 3D boron in relation to the exfoliation mechanism of 2D sheets. Electron-transparent samples were prepared from bulk samples using a focused ion beam (FIB) and imaged under TEM and STEM. Incoherent HAADF STEM, with collection angle β > 60 mrad, typically provides the best separation between atomic columns due to Z-contrast and electron channeling. However, this imaging mode does not provide adequate contrast to distinguish

between the non-aligned low Z atomic columns in boron, as illustrated in Supplementary Fig. 10. We, therefore, resorted to center bright-field (C-BF) STEM−a coherent imaging method relying only on the signal from the center of the convergent beam electron diffraction disk (collection angle β = 0−15 mrad). C-BF displays contrast reversal similar to conventional phase-contrast TEM[39]; by defocusing the electron probe such that the atomic columns appear with bright contrast, we show that all the batches of 3D material have the same β-rhombohedral phase (a representative C-BF image is shown Fig. 2a). This phase of boron has a rhombohedron unit cell consisting of 105 atoms with $R\bar{3}m$ symmetry, lattice parameter a = b= c =10.2 Å, α = β = γ = 65.1° [40], and is internally comprised of icosahedral subunits joined through inter-icosahedral covalent bonds (Fig. 2c). The structure can be visualized as twenty units of $B_{12}$ icosahedrons located at each vertices (eight in total) and middle of each edge (twelve in total) of the unit cell (Fig. 2d), and two $B_{28}$ triple-fused icosahedra flanking a single boron atom centrally positioned in the middle of the rhombohedron (Fig. 2e). As the phases of all starting 3D materials we investigated, regardless of the source, are the same and icosahedral-based, determining other structural/compositional differences is therefore crucial to understand their different propensity for exfoliation.

**Compositional/structural variations between 3D & 2D boron**
To provide additional insights into the crystallographic relation between the structure/composition of the 3D and exfoliated 2D sheets, we next conducted electron energy loss spectroscopy (EELS), X-ray diffraction (XRD) and Raman analysis before and after the exfoliation. Core-loss EELS spectra show similar boron ionization edge (K-edge = 188 eV) and near-edge signature across all starting 3D material (Fig. 2b), suggesting similar elemental composition and bonding. However, for the exfoliated boron, the presence of oxygen impurities is evidenced by the near edge structure at 194 eV[41], and is consistent with previous findings[17,25] (see Supplementary Fig. 11 for further

elemental analysis). XRD results shown in Fig. 2f, confirm the β-rhombohedral structure of the staring material and confirm that the phase of the exfoliated boron does not change after the exfoliation. As the grain diameters of the samples are above values that would influence X-ray kinematical scattering[42], XRD peak broadening can be attributed to the presence of defects in the material. The calculated full width at half maximum (FWHM) of the (111) peak for Samples A, B and C are 0.15°, 0.19° and 0.31°, respectively, and this broadening trend of FWHM confirms the defect analysis from light and electron microscopy imaging. Raman spectroscopy shows that the phonon modes in all the source 3D materials—Samples A, B, and C—are similar, further emphasizing their crystallographic similarity. The boron nanosheets mostly retain the overall Raman profile compared to that of 3D material, but exhibit peak shifts and changes in their intensities, as well as the appearances of new Raman modes (Fig. 2g) indicating preferential orientations during exfoliation.

By comparing our Raman results with previously reported experimental[43,44] and theoretically-calculated[45] Raman-active modes of β-rhombohedral boron (summarized in Table 1), we suggest that the emergence of the Raman peaks at 342 and 430 $cm^{-1}$ in the case of 2D boron nanosheets results from the bond-breaking in both $B_{28}$ and $B_{12}$ icosahedral clusters[43,45]. Furthermore, the complete disappearance of the peak at ~980 $cm^{-1}$ indicates the loss of the inter-icosahedral symmetry, and the shifting of the 1211 $cm^{-1}$ peak to 1140 $cm^{-1}$ may likewise arise from the breaking of the rhombohedral symmetry, which alters the polarization of the central boron atom in the unit cell[43]. We note that the ~309 $cm^{-1}$ peak, and particularly the ~980 $cm^{-1}$ peak are more pronounced in Sample A than in Samples B and C, which is likely due to the presence of planar defects in Samples B and C, as we show below.

## Crystal phase and plane orientation of 2D boron

The changes in the phonon modes of the exfoliated sheets are intriguing as they provide a clue to the exfoliation mechanism based on the broken β-rhombohedral symmetry found in the original crystals. To determine the exact phase and orientation of the LPE-exfoliated nanosheets, we conducted atomic-resolved STEM and matched it with multi-slice simulations. Notably, for rhombohedral and other non-orthogonal crystals, the projected image of its planes stacked at an angle differs depending on the number of planes (up to the thickness where the stack becomes periodic), implying that for thin samples, the thickness can be determined directly from the atomic arrangement in the image. This is demonstrated in Fig. 3, where the atomic models show the image of the projected {100} plane(s) to change depending on whether a single unit cell layer (Fig. 3a) or multilayers of the unit cell (Fig. 3b) are being imaged.

Crucially, even small angle tilts can significantly influence the resulting projected images for icosahedral-based 2D boron (see Supplementary Fig. 13)—which may explain the different phases previously described in the literature. This dependence contrasts with the case for simpler vdW-based nanosheets (see Supplementary Fig. 14 for the comparison), in which small tilts do not alter STEM projection significantly. Figure 3c presents the STEM image of the exfoliated nanosheet, while Fig. 3d–f shows the equivalent simulated projected images of 2, 4, and 10 layers of the {001} plane, respectively. By matching the atomic column brightness and positions in the experimental image in Fig. 3b with the simulated 4 {001} layers image in Fig. 3e, we demonstrate here that: (1) the liquid-phase exfoliated boron sheets planes are of the {001} family, and (2) boron can be exfoliated down to at least 4 unit cells of icosahedral planes, implying thickness of 4.38 nm.

## Atomic-scale STEM characterization of β-rhombohedral boron

In Fig. 4, we show how the atomic-resolved projections of β-rhombohedral boron under different orientations may lead to unintentional structural misinterpretation of 2D boron nanosheets reported in the literature. For example, rhombohedral crystals may appear hexagonal when viewed parallel to their two opposing corners. Under specific viewing orientations, the overlap between $B_{28}$ and $B_{12}$ units may appear seemingly as 'ridges' of brighter contrast. Our results provide the means for boron atomic interpretation regardless of the crystal orientation.

In Fig. 4a–c, experimental C-BF STEM images of the three highest symmetry zone-axes with the largest spacing across the icosahedron columns are directly matched to the simulated image and the corresponding atomic model. When viewed along the [001] and [1$\bar{1}$0] orientations, the $B_{12}$ icosahedron columns are aligned directly parallel to the electron beam, becoming clearly distinguishable, and can be used to identify the β-rhombohedral phase. However, in the case of the [110] projection, ¼ of the $B_{12}$ icosahedra overlap directly with the $B_{28}$ triple-fused icosahedra, and the structure appears with tetragonal symmetry rather than rhombohedral. This, without image simulations, may lead to erroneous interpretation as new boron phases[37]. Interestingly, we found that lower symmetry projections—such as [3$\bar{2}$0] and [4$\bar{1}$0]—when rotated about the (001) plane produce the appearance of seemingly layered structures (Fig. 4d, e). As these projections may lead to potential misinterpretation of the exfoliation mechanism of this boron phase to be similar to vdW materials[21], we show here that these seemingly 'alternating layers' do not indicate the existence of layers but are instead due to the overlap projection of alternating planes of $B_{12}$ and $B_{28}$ units (see the models in Fig. 4d, e).

Figure 4f–j shows the projection of β-rhombohedral boron when viewed perpendicular to neither of its three crystallographic axes. Likewise, for the [1$\bar{1}$1], [112], and [11$\bar{2}$] zone-axes, the cluster of atoms formed from the overlap of both $B_{12}$ and $B_{28}$ units appear with tetragonal and almost cubic-like symmetry (Fig. 4f–h). Although individual atomic columns in the atom clusters are not easily resolved even with aberration-corrected STEM, the distinctive shape of each cluster still allows for their zone-axis determination. We draw particular attention to the [3$\bar{2}$1] projection in Fig. 4j; due to the repeating atom clusters in rhombohedral symmetry appearing seemingly as $B_{12}$ icosahedrons, this zone-axis of β-rhombohedral may potentially be mistaken as the less complicated α-rhombohedral phase. Combining experimental image with simulation, we show that the image still represents the β-rhombohedral phase, and the characteristic six smaller atom clusters surrounding each large cluster can be used for its identification.

## Planar defects in β-rhombohedral boron

We next focus on the high densities of extended crystal defects observed in Sample C, and their relationship to the exfoliation mechanism in boron. Figure 5a provides a visual example of the defect structure through bright-field TEM (similar to the images in Supplementary Fig. 3), where nearly parallel streaks of dark contrast are apparent in the bottom right grain, indicating the presence of planar defects.

Further, atomic-resolved STEM imaging of the defects in the [001] zone-axis (Fig. 5b) reveals that they are parallel to one another, take the form of both twin boundaries and stacking faults (SFs), and significantly, have defect planes of {001} (supporting STEM simulations are shown in the Supplementary Fig. 15). The twinned structure involves a mirror plane inserted along only the $B_{12}$ units (illustrated in Fig. 5c and labeled as {001}-I). The SF boundary (labeled as {001}-II) is formed by breaking one of the triple-fused $B_{28}$ icosahedral units into a single $B_{12}$ icosahedron, and the other into a double-fused $B_{21}$ icosahedral unit. The newly formed $B_{12}$ icosahedron then links with a pre-existing $B_{12}$ unit by sharing a distorted triangular face, forming the SF interface. In this way, the local rhombohedral symmetry of the network of $B_{12}$ units is broken through the creation of the SF. The disruption to the triple-fused $B_{28}$ units also impacts the stability of the center boron atom. These factors can also explain the different Raman modes of the LPE boron and Samples B and C compared to the more pristine Sample A shown in Fig. 2g.

**Table 1 | Raman modes in β-rhombohedral boron from literature[43], and the measured Raman frequencies of 3D Samples A, B, C, and 2D boron**

| Raman peaks region (cm⁻¹) | Raman-active vibrations | Vibrational modes | Ref. 43. | This work | | | |
|---|---|---|---|---|---|---|---|
| | | | | Sample A | Sample B | Sample C | LPE Boron |
| ~200 –400 | Vibration of the $B_{28}$ clusters | | 219 | 235 | 223 | 210 | 243 |
| | | | 282 | 284 | 284 | | |
| | | | 309 | 314 | 314 | 303 | 309 |
| | | | | | | | **342** |
| | | $A_{1g} + E_g$ | 357 | 357 | 358 | 352 | |
| | | | 376 | | 387 | 384 | 378 |
| | | | 391 | | | | |
| ~400 –900 | Intra-icosahedral B–B vibrations, where the strong peak at ~800 cm⁻¹ resulting from the breathing mode of $B_{12}$ clusters | | 414 | | | | |
| | | | | | | | **430** |
| | | $A_{1g} + E_g$ | 456 | | 455 | 450 | 461 |
| | | | 480 | 480 | 477 | 475 | |
| | | | 565 | 530 | 551 | 545 | |
| | | | 594 | 589 | 595 | | 572 |
| | | $A_{1g}$ | 630 | 623 | 624 | 616 | 630 |
| | | | 685 | 683 | 680 | 670 | 672 |
| | | | | 736 | | | 729 |
| | | | 773 | 772 | | 762 | |
| | | $A_{1g}$ | 813 | 813 | 808 | 797 | 785 |
| | | | | | 856 | 845 | 834 |
| | | $A_{1g} + E_g$ | 885 | 882 | 890 | 884 | 895 |
| ~950 –1150 | Inter-icosahedral B-B vibrations | $A_{1g} + E_g$ | 987 | 985 | 985 | 975 | |
| | | | | 1046 | | | |
| | | $A_{1g} + E_g$ | 1097 | 1075 | 1081 | 1075 | 1055 |
| | | | | 1104 | 1106 | 1105 | |
| ~1220 | Stretching vibration of the single B atom in the center of the unit cell parallel to the crystallographic c-axis | $A_{1g} + E_g$ | 1217 | 1230 | 1224 | 1211 | 1140 |

* New Raman peaks that are absent in the original 3D samples, but appear in exfoliated boron are shown in bold.

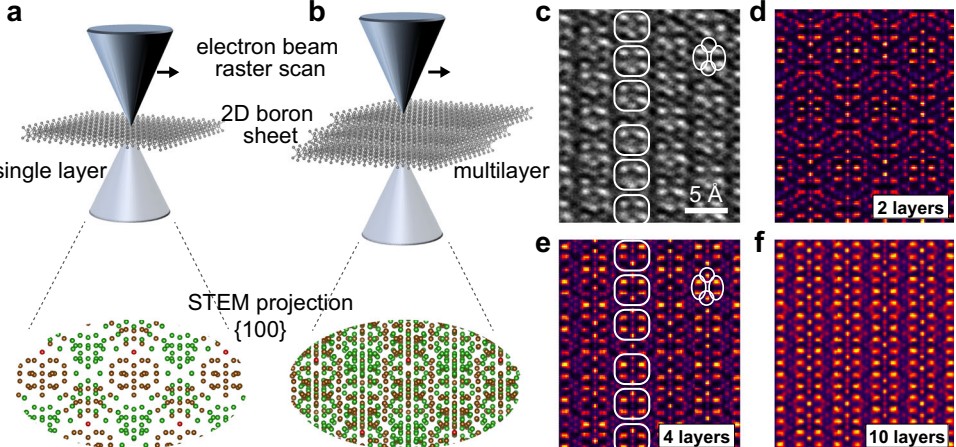

**Fig. 3 | Atomic structure of liquid-phase exfoliation (LPE) derived boron.** Schematic illustrations showing the differences in the STEM projection for boron nanosheets derived from {001} icosahedral planes with (**a**) 1 layer, and (**b**) 4 layers. **c** Atomic-resolved C-BF STEM of an exfoliated nanosheet. **d**–**f** STEM simulations with probe direction perpendicular to the {001} plane of β-rhombohedral boron with 2, 4, and 10 layers of the {001} plane, respectively. The same atomic patterns in the experimental image in c and the simulated image in e are outlined to guide the eye.

It can be observed that planar defects in Sample B and C are always along the {001} plane and parallel to one another, as shown in Fig. 5b and Supplementary Fig. 16. The growth of a crystal occurs along the lowest surface energy facet to minimize free energy; for β-rhombohedral boron, {001} planes have with the lowest surface energy[46], which likely explains why defects are predominantly along the {001} planes. Taken together, our results indicate that planar defects in β-rhombohedral boron play a fundamental role in the exfoliation of these covalently-bonded crystals, where crystal cleavage can occur along these parallel-aligned defect planes during sonication.

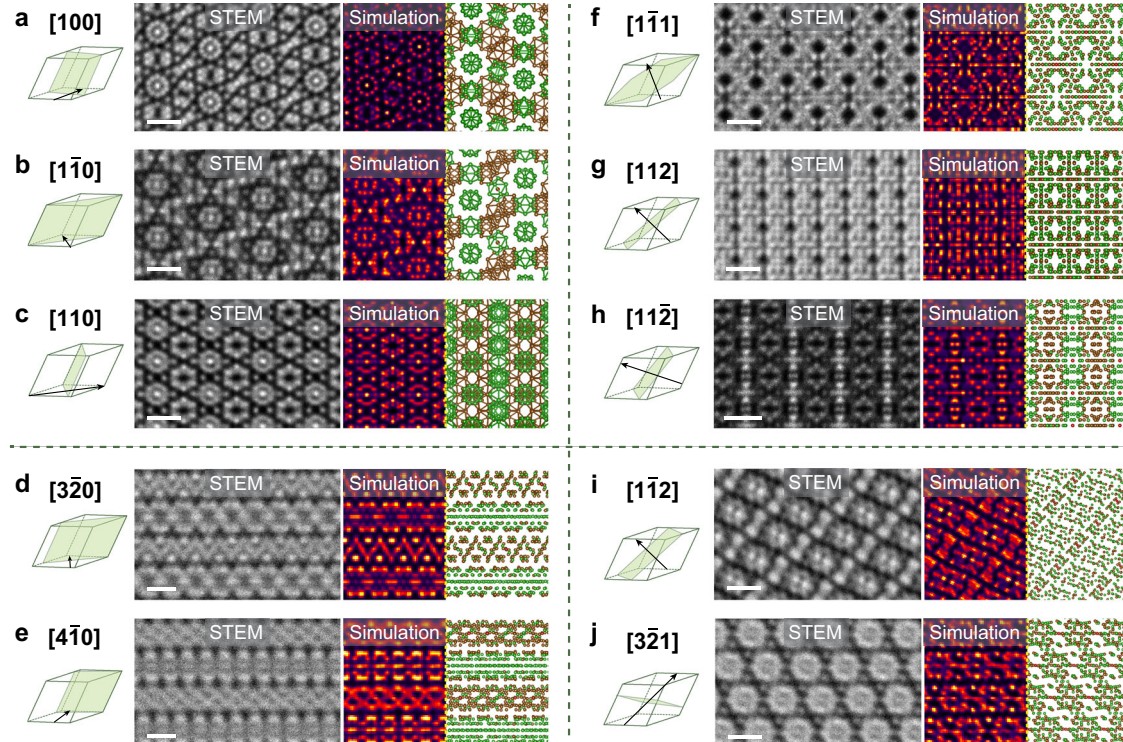

**Fig. 4 | Atomic-structure of 3D β-rhombohedral boron from various zone-axes.** Simulated STEM images and atomic models are shown on the right of the experimental C-BF STEM images (β = 0−15 mrad) images. The $B_{12}$ icosahedrons are depicted in green, $B_{28}$ triple-fused icosahedrons in brown, and the center boron atom in red. The rhombohedrons on the left of the images demonstrate the relation of the electron beam direction (arrow) with the rhombohedral unit cell. (**a**−**c**) show the three projections with the largest spacing between atomic columns where the $B_{12}$ icosahedra columns are aligned parallel to the beam direction. (**d**−**j**) show viewing orientations of lower crystal symmetry. For clarity, bonds are omitted in the models for the lower symmetry projections.

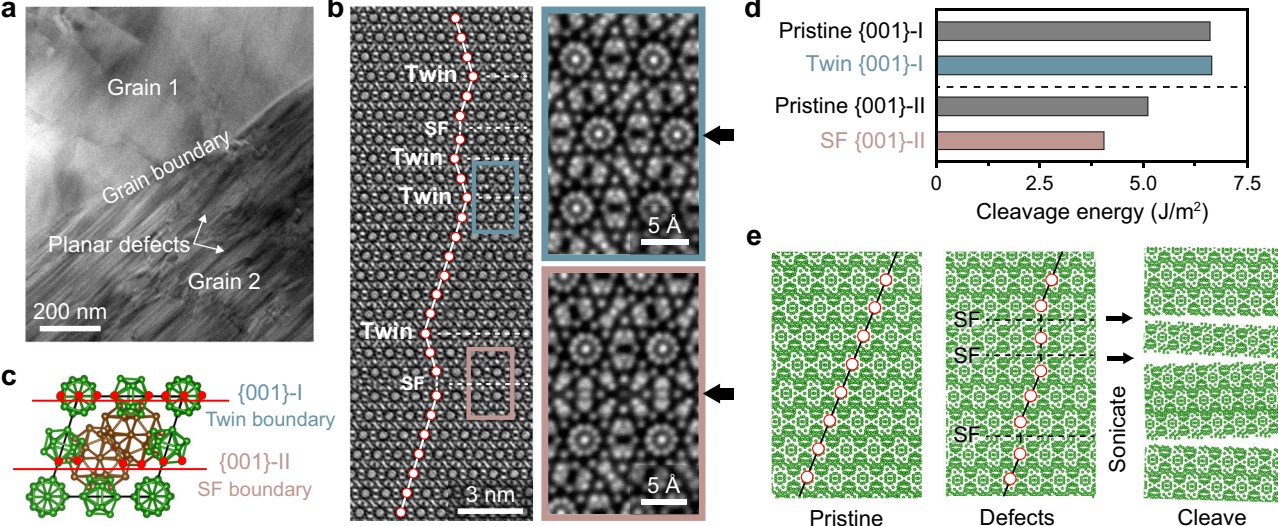

**Fig. 5 | Role of planar defects in the exfoliation mechanism of boron. a** TEM BF image of 3D boron (Sample C) with a high density of planar defects in the bottom right crystal grain 2. **b** C-BF STEM image of β-rhombohedral boron projected along the [001] zone-axis showing a sequence of parallel planar defects, including twins and stacking faults (dashed lines indicate the defect plane). The icosahedral columns are represented as red-ringed circles for easier visualization of the symmetry breaking due to the defects. The magnified images of the twin boundary and stacking fault (SF), boxed in light blue and red, respectively, are shown on the right with the faulted plane marked by the black arrows. **c** Model of the unit cell projected in the [001] depicting the atoms (red circles) forming the defect planes. **d** Density functional theory (DFT) calculated energies for the formation of the {001}-I and {001}-II surfaces from the pristine crystal and from the planar defects. **e** Schematic of the defect-mediated exfoliation process (viewed from the [1$\bar{1}$0] orientation). The icosahedral columns are represented as red-ringed circles, stacking faults as dashed lines, and symmetry breaking of the crystal as solid lines. The black arrows represent the cleaving of the crystal from the stacking faults.

Significantly, this differs from the case of vdW crystals, in which planar defects are not required for exfoliation due to the weaker vdW interactions between layers.

To verify the role of planar defects in the exfoliation of β-rhombohedral boron, we employed DFT[47,48] to calculate the cleavage energies of the {001}-I and {001}-II surfaces from both the pristine crystal and their respective planar defects. Cleavage energy is defined as the energy per unit area required to separate slabs of the same material to form two free surfaces and can be seen simply as twice the surface energy of the cleaved slab[49]. As a starting point for pristine boron calculations, we show that the {001}-II cleavage energy is the lowest at $4.39J/m^2$ and {001}-I surface energy is the highest at $5.70J/m^2$, which is in agreement with the previous report by Hayami et al.[46]. We next calculate the differences in the cleavage energies when cleaving boron from a planar defect as opposed to the pristine boron lattice (Fig. 5d). The presence of an SF further reduces the energy required to form the {001}-II surface to $3.48J/m^2$, which is a significant reduction of 21%. In contrast, we did not observe as significant a difference in cleavage energy from twin boundaries ($5.73 vs 5.70J/m^2$). Our combined DFT and STEM results thus show that stacking faults likely facilitate the exfoliation process of the boron nanosheets from the 3D boron, with crystal cleavage being favored from these faulted planes. Essentially, the presence of these parallelly aligned defect planes produces extrinsic anisotropy in an otherwise isotropic material.

## Discussion

The cleavage preferences along specific defect planes—and the observation that only grains from selected samples contain these defects—can explain the observed nanosheet morphologies of the sonicated products from Samples B and C and the lack of any from Sample A. If a region of crystal is bounded by two parallel planar defects with lower cleavage energy (such as stacking faults, as in the case of β-rhombohedral boron depicted in Fig. 5e, and experimentally shown in Supplementary Figs. 16c and d), a flat nanosheet with uniform thickness will be formed after sonication—with the absolute thickness depending on the separation distance between the two planar defects. Such exfoliation mechanism may also apply to and explain the high aspect ratios of 2D sheets produced from other non-vdW materials, where it is known that planar defects could exist[8,11,13,16]. If the planar defect runs above a grain boundary that is not parallel to the defect, or if the crystal breaks between the defect and {001}-II plane not parallel to it, the exfoliated nanoplatelet will have wedged morphology, as we sometimes observe. The 3D nanoparticles observed among the LPE products likely resulted from the cleaving of the crystal along grain boundaries without specific orientations. This contrasts the distinct exfoliated planes of vdW materials, which follow the vdW layer orientation. From AFM measurements of exfoliated boron (Supplementary Fig. 7e), we do not observe thickness dependence of the nanosheet area, unlike the case of vdW materials, as reported by Backes et al.[50]. This further suggests the exfoliation mechanism of 3D nonlayered materials such as boron is different from vdW materials. Depending on the orientation and densities of planar defects formed in non-vdW materials, different orientations and distributions of nanosheets could be obtained. It is also conceivable that other materials which can demonstrate even larger reduction of cleavage energies from their inherent planar defects, could result in a greater yield of nanosheets.

In summary, by studying the atomic structure of the 2D boron nanosheets and their parental 3D material, we show that the atomic structural morphology in non-vdW crystals plays a significant role in their 2D material exfoliation capabilities. The formation of nanosheets is promoted in β-rhombohedral 3D boron samples with a high density of stacking faults and smaller grains, as opposed to defect-free boron samples with larger grains. Using C-BF STEM, we provide the groundwork for the imaging and interpreting 2D and 3D boron of

various zone-axes and thicknesses, and show that the exfoliated sheets are formed from the {001} planes of the β-rhombohedral phase. The high-density defect planes in the 3D crystal were found to be along the same crystallographic orientation as the exfoliated sheets and parallel to one another. Moreover, our theoretical calculations show lower energy required to cleave the crystal from the stacking fault boundary. Together, our results show that planar defects could be key to an engineerable pathway for the exfoliation of low-dimensional sheets from boron and from a wider range of covalently bonded materials.

## Methods

### Liquid-phase exfoliation of boron

Three different starting bulk boron materials were used for liquid phase exfoliation: (1) Sigma-Aldrich Boron (crystalline, 99.7%, trace metals basis), (2) Alfa Aesar Boron pieces (crystalline, 99.4% metals basis) and (3) Yamanaka Advanced Materials UHP Boron powder; 20 mg boron bulk was added to 10 mL of acetone, and bath-sonicated for 16 h using a Elma Ultrasonics TI-H10 at 35 kHz, power of 200 W and at temperature of 300 K.

### Raman spectroscopy and AFM characterization

For grain analysis, bulk boron crystals were polished using diamond lapping films (final film grade of $0.1\,\mu m$), and viewed under a polarized light microscope (Olympus DSX1000) with the polarizer angle set between 85°–90°. For Raman characterization, the sonicated boron samples were drop-casted to a silicon substrate and annealed at 500 °C under high vacuum ($10^{-7} \cdot 10^{-8}$ Pa) for 6 h. Raman measurements of both 3D boton nanocrystals and 2D boron nanosheets were performed using a HORIBA Scientific LabRAM HR Evolution Raman Spectrometer, with excitation laser 488 nm (Lexel SHG 95 argon ion) and the spectra recorded in backscattered geometry. Following exfoliation guidelines from Backes et al.[51], centrifugation was conducted at 3421 g for 30 min. The supernatant were decanted, then spin-coated onto a Si wafer with 300 nm of native oxide to prevent agglomeration. The wafers were further annealed under high vacuum ($10^{-8}$ mbar) for 6 h to remove solvent patches. AFM measurements of LPE boron was conducted with a Bruker Dimension Fastscan system, and data further processed with the NanoScope Analysis software. XRD for the bulk and exfoliated boron was conducted using a Bruker D8 Advance, and VESTA was used to simulate the diffraction pattern.

### Atomic-resolved STEM characterization and simulation

Electron transparent samples were prepared from bulk boron using the in-situ lamella lift-off method in a FEI Versa 3D DualBeam system (final cleaning step at 2 kV, 27 pA), while exfoliated boron for TEM imaging were prepared through drop-casting the sonicated medium on Quantifoil holey carbon grids, which were annealed at 500 °C under vacuum conditions of $10^{-7} \cdot 10^{-8}$ Pa for 6 h. Atomically-resolved STEM images were acquired with a JEOL JEM-ARM200F equipped with a cold field emission gun and a fifth-order ASCOR aberration corrector operated at 200 kV, and ~31 mrad probe convergence angle $\alpha$. For characterizing the boron atomic structure, center bright-field images were captured with a circular detector (collection semi-angle $\beta$: ~0–15 mrad, performed at a camera length of 20 cm), and defocused to the contrast regime where the atomic columns appear bright, and vacuum dark. EELS data were acquired at 80 kV (collection semi-angle: ~77 mrad). To aid in STEM image simulations, the atomic models were constructed and rotated using the Atomsk software package[52], cropped and visualized through VESTA[53]. Multislice STEM simulations were performed using the Prismatic simulation code (pixel size = 0.02 Å, potential bound = 2 Å, accelerating voltage = 200 kV, $\alpha$ = 31 mrad, $\beta$ = 0-15 mrad)[54].

### DFT calculations

The DFT calculations for the surface energy was performed using the sufficiently constrained and appropriately normed (SCAN) meta-GGA

exchange correlation functional using a projector augmented-wave (PAW) potentials to describe the B wave functions, as implemented in the Vienna Ab Initio Simulation Package (VASP) code[47,48]. In the PAW formalism for B, the electrons that are treated explicitly are $1s^2 2p^1$. A plane-wave basis set with an energy cut off of 520 eV and $2 \times 2 \times 2$ Γ-centered Monkhorst-Pack $k$- point meshes were used for the calculation of the primitive cell of 3D B (105 atoms f.u. and $R\bar{3}m$ space group). For the structures containing the stacking fault and twin, as well as for the {001}-I, {001}-II, and stacking fault boundary surfaces, larger supercells were constructed and the $k$-point mesh were adjusted to obtain the same sampling of the first Brillouin zone. The {001}-I, {001}-II and stacking fault boundary surfaces were constructed as slabs with ~1.2 nm vacuum distance to prevent interactions between the surfaces (see Supplementary Fig. 17).

The cleavage energies of the {001}-I and {001}-II separated from the pristine, twinned, and stacking fault structures in 3D crystals were calculated using the following equation:

$$Cleavage\ energy = \frac{E_{S1} + E_{S2} - E_{crystal}}{A},\qquad(1)$$

where $E_{S1}$, $E_{S2}$, and $E_{crystal}$ are the energies of slab 1, slab 2, and the crystal, respectively; and $A$ is the surface area of the cleaved {001} interface surface.

## Data availability
The EELS, XRD, and Raman data generated in this study are provided in the Source Data file. Any additional data that supports the findings of this study, and models used for simulation, calculation, and visualization are available from the corresponding authors upon request. Source data are provided in this paper.

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

## Acknowledgements

We acknowledge support by Agency for Science, Technology and Research (A*STAR) Singapore under its Advanced Manufacturing and Engineering (AME) Programmatic grant (Award A18A9b0060, SGa), and support by Ministry of Education (MOE) Academic Research Fund (AcRF) Tier 2 (MOE-T2EP50221-0018, SGa, and MOE-T2EP50220-0016, SGr). ESS and SGa acknowledge support from a United Arab Emirates University–Asian Universities Alliance (UAEU–AUA) Joint Research Project (Grant Number 31R196). J-YC and YY were supported by the National Research Foundation, Singapore under its RIE Roles for SGUnited Jobs - NUS Tranche 1 Project (NRF-MP-2020-0004). The authors also acknowledge the Facility for Analysis, Characterization, Testing and Simulation, Nanyang Technological University, Singapore, for use of their electron microscopy facilities and Materials Science Shared Facilities, National University of Singapore, for access to XRD, Raman and TEM, and Dr Wu Jiang from Oxford Instruments for EBSD measurements.

## Author contributions

S.Ga. and S.Gr. conceptualized (with initial input from ESS) and supervised the project. Y.Y., C.J. and P.R. synthesized the L.P.E. boron. Y.Y. and C.J. performed the X.R.D., A.F.M. and Raman measurements/analysis. J.-Y.C. prepared the TEM samples, conducted the polarized light and TEM/STEM measurements/simulations, and generated the models for simulations/calculations. T.P.M. performed the DFT calculations and analysis under the supervision of PC. All authors contributed to the writing of the manuscript.

## Competing interests

The authors declare no competing interests.
