## [Peer Review File · Nature Communications]

Structure and exfoliation mechanism of two-dimensional boron nanosheetsREVIEWER COMMENTS

Reviewer #1 (Remarks to the Author):

The manuscript attempts to elucidate the liquid phase exfoliation mechanism of 3D non-layered, non-vdW materials into 2D nanosheets, using 3D-boron as a case study. The authors report the liquid-phase exfoliation of 3D boron into 2D nanosheets and propose a mechanism based on planar defects in the 3D lattice. The authors claimed that the planar defects in the starting 3D material promote exfoliation into 2D sheets. The authors used STEM, and Raman spectroscopy to experimentally characterize the orientation and structure of starting 3D-boron and liquid-produced 2D nanosheets. DFT calculations were also performed to affirm that the observed planar defects possess the lowest cleavage energy planes-suggesting defects mediate the exfoliation mechanism. However, the reviewer believes that the current presentation of results is deemed insufficient to substantiate the proposed exfoliation mechanism. In its current form, the manuscript is not recommended for publication in the Journal of Nature Comm. To improve the quality of the manuscript, the following suggestions are provided:

1. The authors evaluated the grain size of bulk boron sourced from three different companies using SEM and LAADF STEM imaging. The data cannot be believed as both are selective imaging techniques, and the sourced 3D-boron particles consist of a broad distribution of the size of particles. Using these techniques to comment on the grain size is not reliable, as the surface of each particle in each sample can look different in imaging. Also, this analysis can be affected by an observational bias, therefore it is recommended to include XRD data and calculate the average crystallite size for each sample.

2. The authors did the LPE of 3D-boron in solvent acetone, and bath sonicated for 16h. The exfoliation details are incomplete such as the temperature of the bath which will affect the exfoliation procedure. Further, it is well known that an exfoliation process results in a stock dispersion consisting of unexfoliated (bulk particles) and exfoliated products (2D-like). Therefore, centrifuging the dispersion at a particular g-force is necessary for a certain period to remove the unexfoliated bulk-like particles from the dispersion[<https://doi.org/10.1021/acs.chemmater.6b03335>]. This is necessary before the characterization of dispersion and to comment on the yield (Figure S3(a)). The SEM image in SI, Figure S3 (b) clearly shows bulk nanoparticles rather than 2D-like nanosheets in the dispersion may be due to the above reason.

3. As explained by others [<https://doi.org/10.1021/acs.chemmater.6b03335>, <https://doi.org/10.1002/adma.202202164>], it is critically important to measure the dimensions of the exfoliated nanosheets as parameters such as L, W and t of nanosheets which has proved very important for understanding the exfoliation mechanism of 2D materials. Such measurements become more important while dealing with LPE of 3D materials, such as boron. We ask authors to perform statistical AFM analysis to calculate the average L, W, and t with acceptable errors from

each dispersion to verify that the nanosheets of boron were produced by exfoliation of 3D-boron in Acetone. Also, it has been found that for LPE of 2D materials, these aspect ratios can be related to the models of the exfoliation mechanism [<https://doi.org/10.1021/acsnano.9b02234>, <https://doi.org/10.1021/acsnano.0c03916>]. We suggest authors do this analysis to prove that the exfoliation mechanism of 3D nonlayered materials is different from 2D materials.

4. We suggest authors do XRD measurements on the exfoliated samples of all three dispersions to support the Raman results. This is particularly important for identifying the diffraction peak shifts related to defects.

5. In addition to STEM, the authors should include XRD data to study the crystal defects. This would mitigate potential observational bias associated with selective imaging techniques.

6. In Figure 5 in the manuscript, the TEM images of 3D-boron planar defects are from sample B or sample C? Do both samples B and C have similar defect densities? Also, are the planar defects in both samples B and C have the same orientation? Then, why is the exfoliation yield from samples B and C different (from the color in Figure S3 (a), it seems sample C has a high exfoliation yield as compared to sample B)?

7. The manuscript should address why the stacking faults in the 3D-boron are limited to only {001} planes. Providing an explanation or exploring the implications of this limitation would enhance the manuscript's depth.

8. Why multiple cleavage planes have been observed in the exfoliation of 3D materials [<https://doi.org/10.1002/adma.202202164>]?

Reviewer #2 (Remarks to the Author):

This paper reports a plausible mechanism via which 2D nanosheets can be produced during liquid phase exfoliation of a 'non-layered' material, via planar defects aligned along the observed LPE nanosheet cleavage direction. The imaging is impressive, the image interpretation aided by use of simulations is convincing, and the manuscript is very well written and presented. Complementary spectroscopic data and surface energy calculations support the hypothesis. While it is not clear to me how well this explanation would translate for similar materials I think it is an interesting result, and would be of interest to readers of nature communications.

I have some concerns over the main result however. An attempt to roughly quantify the frequency of the twin boundaries and stacking faults in the 3 materials would be helpful if possible. From my reading it seems that the twin boundary would barely contribute to promoting cleavage, and it is the stacking fault which would dominate. However only a single stacking fault is observed, and no closely spaced stacking faults which would promote exfoliation of thin nanosheets. It could be the case that it is a combination of the stacking faults and the presumably minimal cleavage energy along the 001-ii plane which promotes generation of thin sheets? Are there enough stacking faults relative to the observed quantity of generated nanosheets to support the idea that the defects are playing an important role? The visualisation of planar defects in fig 5 and the SI and lack of the additional raman peaks do not necessarily show the crucial stacking fault, and the absence of them

in sample A is not demonstrated as far as I can see.

Reviewer #3 (Remarks to the Author):

Recommendation: Major revision

Manuscript ID: NCOMMS-23-63421

Comments:

In this manuscript, the authors aim to reveal the underlying formation mechanism of 2D structures exfoliated from non-layered materials. This is a really interesting research topic. Especially, it has been proven that 3D boron, as a typical non-layered material, can be used to produce few-layer boron sheets in large quantities by liquid-phase exfoliation method. However, in my opinion, the field still does not have any reasonable mechanism to explain the formation of two-dimensional boron. This has also led to some experimental reports of similar preparation methods but completely inconsistent characterization and analysis results. Through comparative experiments, the authors found that atomic structural morphology in non-vdW crystals, such as high density of stacking faults and small grains, can promote the formation of 2D structure. Through the combination of theory and experiment, the author also proves that planar defects could be key to an engineerable pathway for the exfoliation of low-dimensional sheets from boron. Although I think this topic is of great significance in the field of liquid-phase exfoliation of two-dimensional boron, this manuscript contains many ambiguous points. I think that after addressing the below concerns the paper can certainly be considered again for publication in Nature Communications. Detailed comments are listed below.

1. In Abstract, the authors state that “We show that planar defects in the starting 3D material promote the exfoliation of 2D sheets, ..., and density functional theory calculations.”. I think the description is a bit too general and should be made more specific to mention the boron material system. Similar problems occur in the following descriptions: Taken together, our results indicate that ... due to the weaker vdW interactions between layers.
2. Ultrasonic power is an important index in the liquid phase exfoliation method. The author needs to mention the specific value of the ultrasonic power used in the experiment in the manuscript.
3. In The crystal phase of 3D boron, please provide XRD data for 3D boron and exfoliated 2D boron. In addition, the corresponding discussions should be also added in this section.
4. In Surface passivation of boron sheets with oxygen, I think the authors should provide the corresponding B 1s spectrum of 3D boron and compare it with the 2D counterpart to understand the reason for the formation of B-O in BxO.
5. The authors state that LPE of boron was sometimes found to yield ‘sheet-like’ layered products with irregular shapes. I would like to know whether the ‘sheet-like’ layered products can have impact on the Raman results for the exfoliated 2D boron. In other words, can the author provide Raman spectra of the ‘sheet-like’ layered products?
6. As we all know, the choice of the exfoliating solvents is crucial for the generation of nanosheets. Why did the author only choose to use acetone as the exfoliating solvent? For sample A, can nanosheets be produced in other ultrasonic solvents? If so, the conclusion of this manuscript

seems to be less rigorous. Therefore, I hope that the author can supplement the experimental results of sample A-C in other types of ultrasonic solvents.

7. In Introduction, the authors state that “boron nanosheets have drawn particular interest due to their potential for a range of applications, including sensors, photoelectronics, energy storage, drug delivery, bioimaging, and catalysts”. The authors should cite some related experimental breakthroughs on the application of 2D boron. For example: Nano Energy, 2022, 97, 107189; Nano Research Energy 2023, 2, e9120051; Nano Res. 2021, 14, 2337-2344; J. Mater. Chem. A 2021, 9, 13100-13108; Nano Res. 2022, 15, 2537-2544; Angew. Chem. Int. Ed. 2020, 132, 10911-10917; ACS Applied Materials & Interfaces, 2023, 15(11), 14566–14574; Nano Research, 2023, 16, 5826-5833.

8. Please ensure that the order of references corresponds to the description in the manuscript. The format of the reference needs to be in accordance with the format required by Nature Communications. There are a lot of format errors in the manuscript, for example: references 2, 7, 8 and 14.

NCOMMS-23-63421 Response to the reviewer's comments

We thank the reviewers for their comments. Their questions and comments are addressed point-by-point in full, as presented below.

We have added another author, Ms. Chithralekha Joseph, a PhD student who helped performing additional support experiments suggested by the referees.

Reviewer 1

Comment 1. The authors evaluated the grain size of bulk boron sourced from three different companies using SEM and LAADF STEM imaging. The data cannot be believed as both are selective imaging techniques, and the sourced 3D-boron particles consist of a broad distribution of the size of particles. Using these techniques to comment on the grain size is not reliable, as the surface of each particle in each sample can look different in imaging. Also, this analysis can be affected by an observational bias, therefore it is recommended to include XRD data and calculate the average crystallite size for each sample.

Reply 1: We thank the Reviewer for the comment. In the revised manuscript (revised **Figure 2**) we now show X-ray diffraction (XRD) data for all investigated 3D samples, which corroborate our results and confirm β -rhombohedral phase of boron. We respectfully point out that the mean crystallite sizes for our samples are in the micrometer scale, whereas the method of estimating mean crystallite sizes through XRD is limited to sizes of $< 0.1\text{--}0.2\ \mu\text{m}$, and measurable peak broadening is noticeable below $\sim 50\ \text{nm}$ (see <https://doi.org/10.1021/acsnano.9b05157>).

Therefore, following the spirit of the reviewer's suggestion, we additionally performed statistically significant analysis of grain sizes using large field-of-view polarized light microscopy shown in the **Supplementary Fig. S1**. By measuring > 300 grains per sample, we determine the mean crystallite diameters of Samples A, B and C to be $148.2 \pm 78.9\ \mu\text{m}$, $105.3 \pm 30.8\ \mu\text{m}$ and $11.0 \pm 3.7\ \mu\text{m}$, respectively.

Three sentences have been added in **Page 6, Lines 4-12**: "From polarized light microscope images in **Supplementary Material (SM)**, **Error! Reference source not found.** Fig S1a and b [...] as well as TEM/STEM images in **SM, Fig. S1a and S3**."

Comment 2. The authors did the LPE of 3D-boron in solvent acetone, and bath sonicated for 16h. The exfoliation details are incomplete such as the temperature of the bath which will affect the exfoliation procedure. Further, it is well known that an exfoliation process results in a stock dispersion consisting of unexfoliated (bulk particles) and exfoliated products (2D-like). Therefore, centrifuging the dispersion at a particular g-force is necessary for a certain period to remove the unexfoliated bulk-like particles from the dispersion [<https://doi.org/10.1021/acs.chemmater.6b03335>]. This is necessary before the characterization of dispersion and to comment on the yield (Figure S3(a)). The SEM image in SI, Figure S3 (b) clearly shows bulk nanoparticles rather than 2D-like nanosheets in the dispersion may be due to the above reason.

Reply 2: We agree with the Reviewer about the importance of the centrifugation, which was indeed carried out for atomic force microscopy (AFM) analysis shown in **SM, Fig. S7**. The intent of **Figure S4** (previously **S3**) was precisely to demonstrate the Reviewer's point: that breaking the bulk 3D boron through sonication results in not only nanosheets but also wedged platelets and other bulk products. We added the relevant exfoliation details in the Methods section of the main text and Supplementary Information, including the temperature of the bath (300 K), sonication power (200 W), and

centrifugation details, as well as the well-accepted reference for guidelines on liquid-phase exfoliation (LPE) (<https://doi.org/10.1021/acs.chemmater.6b03335>).

Comment 3. As explained by others [<https://doi.org/10.1021/acs.chemmater.6b03335>, <https://doi.org/10.1002/adma.202202164>], it is critically important to measure the dimensions of the exfoliated nanosheets as parameters such as L, W and t of nanosheets which has proved very important for understanding the exfoliation mechanism of 2D materials. Such measurements become more important while dealing with LPE of 3D materials, such as boron. We ask authors to perform statistical AFM analysis to calculate the average L, W, and t with acceptable errors from each dispersion to verify that the nanosheets of boron were produced by exfoliation of 3D-boron in Acetone.

Reply 3: Following the suggestion of the Reviewer, we performed the statistical AFM measurements from 100 flakes from Sample C. **SM, Fig. S7** now includes representative AFM images of a single and multiple flakes, their height profile and the statistical analysis of the flakes' surface area and thickness. The mean area of the exfoliated flakes is $0.02 \mu\text{m}^2$, with an average thickness of 3.9 nm. These values match our scanning transmission electron microscopy (STEM) analysis in **Figure 1c** and **Figure 3c**. Care was also taken to only measure true flakes, and not solvent residues (as already shown in the **SM Figs. S8 and S9**). For the case of Samples A and B, the yield is insufficient for statistical analysis.

These results are in agreement with past reports focusing on LPE of boron (References 17-25), with a number of them showing statistical AFM analysis. For example, both the thicknesses and area of the flakes from Sample C are similar to the ones from Reference 17, which studied over 200 LPE boron flakes. Our manuscript does not aim to use AFM statistics to demonstrate the production of boron flakes through LPE, as this has already been well established, even according to <https://doi.org/10.1002/adma.202202164>. Instead, our goal is to gain a deeper understanding of the mechanism behind the exfoliation of this isotropic material.

To address this comment, we added at sentence to **Page 7, line 2** on AFM: "AFM imaging of 100 flakes likewise demonstrates their mean thickness to be 3.9 nm (**SM, Fig. S7**)."

Comment 4. Also, it has been found that for LPE of 2D materials, these aspect ratios can be related to the models of the exfoliation mechanism [<https://doi.org/10.1021/acsnano.9b02234>, <https://doi.org/10.1021/acsnano.0c03916>]. We suggest authors do this analysis to prove that the exfoliation mechanism of 3D nonlayered materials is different from 2D materials.

Reply 4: The reference mentioned by the Reviewer [<https://doi.org/10.1021/acsnano.9b02234>] correlates the aspect ratios of various LPE-exfoliated 2D materials to material-specific parameters, such as in-plane/out-of-plane Young's moduli. From the new **SM Figure S7e**, we do not observe thickness dependence of the nanosheet area, unlike the case of LPE of 2D materials reported in the reference. This result indicates that the exfoliation mechanism of 3D nonlayered materials such as boron is indeed different from 2D materials.

We added a comment and reference [<https://doi.org/10.1021/acsnano.9b02234>] in **Page 17, Lines 9-13**: "From AFM measurements of exfoliated boron (**SM, Fig. S7e**), we do not observe thickness dependence [...] the exfoliation mechanism of 3D nonlayered materials such as boron is different from vdW materials."

Comment 5. We suggest authors do XRD measurements on the exfoliated samples of all three dispersions to support the Raman results. This is particularly important for identifying the diffraction peak shifts related to defects.

&

In addition to STEM, the authors should include XRD data to study the crystal defects. This would mitigate potential observational bias associated with selective imaging techniques.

Reply 5: We thank the reviewer for the suggestions. In the revised **Figure 2** of the main text, we now show XRD patterns for Samples A, B and C, as well as the sonicated products from Sample C, which further support our conclusions. No significant changes were observed in the XRD patterns of Sample C after sonication, indicating no phase change occurred during the process, consistent with our Raman measurements.

As already discussed in Reply 1, the grain sizes are above values that would influence X-ray kinematical scattering and thus, XRD peak broadening can be attributed to the presence of defects in the material. The calculated full width at half maximum (FWHM) of the (111) peak for Samples A, B and C increases from 0.15 to 0.19 and 0.31, respectively, which suggests the following trend of defect density (DD): $DD(\text{Sample A}) < DD(\text{Sample B}) < DD(\text{Sample C})$. These results also match the qualitative observations from polarized light microscopy (**SM, Fig. S2**), electron backscattered diffraction (EBSD) and STEM (**SM, Figs. S1 and S3**).

Comment 6. In Figure 5 in the manuscript, the TEM images of 3D-boron planar defects are from sample B or sample C? Do both samples B and C have similar defect densities? Also, are the planar defects in both samples B and C have the same orientation? Then, why is the exfoliation yield from samples B and C different (from the color in Figure S3 (a), it seems sample C has a high exfoliation yield as compared to sample B)?

Reply 6: **Figure 5** shows STEM images from Sample C, which we clarify in the revised manuscript. From XRD results discussed in Reply 5, the defect density in Sample C is higher, explaining why the exfoliation yield in sample C is significantly higher compared to sample B. Planar defects in Sample B have the same (001) orientation, which we now show in **SM, Fig. S16**.

Comment 7. The manuscript should address why the stacking faults in the 3D-boron are limited to only {001} planes. Providing an explanation or exploring the implications of this limitation would enhance the manuscript's depth.

Reply 7: Stacking faults in materials can arise during crystal growth or from plastic deformation. Boron is exceptionally brittle, implying the defects are essentially immobile and hence do not originate from plastic deformation. Hence, planar defects in boron likely originate from stacking mismatch along the crystal growth facet. The growth of a crystal occurs along the lowest surface energy facet to minimize free energy; for β -rhombohedral boron, {001} planes have with the lowest surface energy (<https://doi.org/10.1021/jp065680s>), which likely explains why defects are predominantly along the {001} planes.

A statement has been added to **Page 14, Lines 22-23, and Page 15, Lines 1-3**: "It can be observed that planar defects in Sample B and C are always along the {001} plane [...] explains why defects are predominantly along the {001} planes."

Comment 8. Why multiple cleavage planes have been observed in the exfoliation of 3D materials [<https://doi.org/10.1002/adma.202202164>]?

Reply 8: The lack of in-depth study on the starting bulk material for LPE of non-vdW materials is staggering, considering the large number of published LPE papers. Therefore, it is challenging to comment on this question by the reviewer without detailed structural analyses on the starting materials covered by the review paper <https://doi.org/10.1002/adma.202202164>.

We could however look into past defect studies for comparison. In the hematite exfoliation paper (<https://doi.org/10.1038/s41565-018-0134-y>), the authors suggest the two cleavage planes from corundum Fe_2O_3 are (0001) and (01 $\bar{1}$ 0). From previous studies on corundum crystal structures, planar defects (including stacking faults [https://doi.org/10.1016/0001-6160\(84\)90206-2](https://doi.org/10.1016/0001-6160(84)90206-2), as well as twins <https://doi.org/10.1080/01418619808241934>) can exist along both the basal and prismatic planes, matching the two exfoliated planes proposed by the authors. This is consistent with our conclusions.

Reviewer 2

General Comment: This paper reports a plausible mechanism via which 2D nanosheets can be produced during liquid phase exfoliation of a 'non-layered' material, via planar defects aligned along the observed LPE nanosheet cleavage direction. The imaging is impressive, the image interpretation aided by use of simulations is convincing, and the manuscript is very well written and presented. Complementary spectroscopic data and surface energy calculations support the hypothesis. While it is not clear to me how well this explanation would translate for similar materials I think it is an interesting result, and would be of interest to readers of nature communications.

Reply: We thank the Reviewer for their overall constructive comments and positive feedback on our imaging and analysis.

Comment 1. I have some concerns over the main result however. An attempt to roughly quantify the frequency of the twin boundaries and stacking faults in the 3 materials would be helpful if possible.

Reply 1: The identification and differentiation between twin boundaries and stacking faults is challenging except through transmission electron microscopy (TEM) analysis, which is not feasible for large-scale analysis. However, to address the Reviewer's comment, we qualitatively compared the overall defect density using X-ray diffraction (XRD), large field-of-view direct imaging and electron back-scattered diffraction (EBSD).

In **Figure 2d** we now plot the XRD patterns for all three samples. The calculated full width at half maximum (FWHM) of the first (111) peak for Samples A, B and C are 0.15° , 0.19° and 0.31° , respectively. The grain sizes in all three samples are in the micrometer scale and thus above values that would influence X-ray kinematical scattering. XRD peak broadening can hence be attributed to the presence of defects in the material. The highest FWHM in Sample C indicates the highest density of defects in this sample, consistent with our exfoliation results.

The XRD results are supported by the defect analysis from large field-of-view microscopy imaging (see also the response to Reviewer 1, Comment 5). We conducted polarized light microscopy on the polished surfaces of all three starting materials. Because planar defects change the orientation of a crystal grain, they are observed as parallel streaks across the grains (see **Supplementary Materials (SM), Fig. S1**). Horizontal lines are observed in both Samples A and B; qualitatively however, closely spaced lines are more predominantly seen in Sample B (**SM, Fig. S2**).

For the case of Sample C, the defects are so closely spaced such that light microscopy does not provide sufficient spatial resolution for analysis. We thus turn to EBSD, which offers higher spatial resolution of the changes in crystal orientation resulting from these defects. As seen in the inset in **SM, Fig. S1a**,

streaking can also be observed in Sample C, which is further corroborated with TEM/STEM images in **SM, Fig. S3** from randomly prepared FIB lamella from Sample C.

Comment 2. From my reading it seems that the twin boundary would barely contribute to promoting cleavage, and it is the stacking fault which would dominate. However only a single stacking fault is observed, and no closely spaced stacking faults which would promote exfoliation of thin nanosheets.

&

Are there enough stacking faults relative to the observed quantity of generated nanosheets to support the idea that the defects are playing an important role? The visualisation of planar defects in fig 5 and the SI and lack of the additional Raman peaks do not necessarily show the crucial stacking fault, and the absence of them in sample A is not demonstrated as far as I can see.

Reply 2: We thank the Reviewer for the comment as the new results further support and strengthen our original conclusions. To our knowledge, there is no available broad-based characterization technique that can systematically screen and identify the spacing between stacking faults. Thus, we are providing further STEM examples to illustrate that (1) our image of the stacking fault in the original manuscript is representative, and (2) stacking faults are common enough and closely packed in Sample C to promote exfoliation of thin nanosheets.

In **Figure 5b**, we now replace the image with a larger field-of-view STEM image taken from sample C showing two stacking faults and four twin boundaries. The high magnification images of the defect boundaries have also been retaken at significantly higher resolution. **SM, Figs. S16c–d** now also shows two other examples of closely-spaced stacking faults bounding only two and four layers of the (001) planes. If exfoliated, these would yield theoretical flake thicknesses of 2.19 nm and 4.38 nm, respectively, consistent with our AFM results. Notably, the four-layer example in **SM, Fig. S16d** matches the thickness of the flake shown in **Figure 3** and the AFM thicknesses analysis in **SM, Fig. S7**.

Similar to the reply to Comment 1, broad-based measurements suggest the defect density is the lowest in Sample A. However, as rightly pointed out by the Reviewer, we cannot conclusively state that there are absolutely no stacking faults in Sample A. As such, in **Page 14, line 4**, we removed the hard statement: “[...] high densities of extended crystal defects observed in Samples B and C, ~~but not in Sample A.~~”

Comment 3. It could be the case that it is a combination of the stacking faults and the presumably minimal cleavage energy along the 001-ii plane which promotes generation of thin sheets?

Reply 3: This is a good point, and the combination of the {001}-ii and a stacking fault could indeed lead to thin sheets as well. However, unlike stacking faults which are always parallel to one another, the {001}-ii could exist along any of the three rhombohedral axes. Breaking from a stacking fault and another non-parallel {001}-ii may help to explain why wedged particles were observed.

In **Page 17, Line 5**, a short statement has been added: “[...] or if the crystal breaks between the defect and {001}-II plane not parallel to it, [...]”

Reviewer 3

Comment 1. In Abstract, the authors state that “We show that planar defects in the starting 3D material promote the exfoliation of 2D sheets, ..., and density functional theory calculations.”. I think the description is a bit too general and should be made more specific to mention the boron material system.

Similar problems occur in the following descriptions: Taken together, our results indicate that ... due to the weaker vdW interactions between layers.

Reply 1: We thank the reviewer for this comment. The abstract has been edited to focus on the boron system, with a suggestion that it may also be applicable to other material systems.

Page 2, Line 7: “[...] starting 3D boron material [...]”

Page 15, Line 8: “Taken together, our results suggest that these parallel-aligned planar defects in β -rhombohedral boron play a fundamental role in [...]”.

Comment 2. Ultrasonic power is an important index in the liquid phase exfoliation method. The author needs to mention the specific value of the ultrasonic power used in the experiment in the manuscript.

Reply 2: The Methods section now includes more details, including the sonication power (200 W), the temperature of the bath (300 K), and centrifugation details.

Comment 3. In the crystal phase of 3D boron, please provide XRD data for 3D boron and exfoliated 2D boron. In addition, the corresponding discussions should be also added in this section.

Reply 3: We thank the reviewer for the good advice for XRD measurements on the bulk and exfoliated boron. In **Figure 2**, XRD patterns for Samples A, B and C, as well as the sonicated products from Sample C are now shown. The calculated full width at half maximum (FWHM) of the (111) peak for Samples A, B and C are 0.15, 0.19 and 0.31, respectively, which suggests the following trend of defect density (DD): DD(Sample A) < DD(Sample B) < DD(Sample C).

These results also matches the qualitative observations from: (1) polarized light microscopy (**Supplementary Materials (SM), Fig. S2**), which shows a greater number of closely-spaced horizontal lines (indicating defects) in Sample B compared to Sample A, and (2) electron backscattered diffraction (EBSD) and scanning transmission electron microscope (STEM) (**SM, Figs. S1 and S3**), which shows dense strain-contrast lines in Sample C. Taken together, these results further support our original conclusions.

In **Page 9, Lines 9-16**, the description for XRD analysis has been added: “XRD results shown in **Fig. 2f**, confirm the β -rhombohedral structure [...] broadening trend of FWHM confirms the defect analysis from light and electron microscopy imaging.”

Comment 4. In Surface passivation of boron sheets with oxygen, I think the authors should provide the corresponding B 1s spectrum of 3D boron and compare it with the 2D counterpart to understand the reason for the formation of B-O in B_xO.

Reply 4: We apologize for the confusion due to a lack of labelling. The B1s X-ray photoelectron spectroscopy (XPS) spectra of 3D boron and the 2D counterpart are already shown in the **SM, Fig. S11**, showing no gross-scale bonding changes occur during the exfoliation process. We also note that previous reports have also indicated the presence of the B–O peak in bulk boron, attributed to the native oxide (see <https://doi.org/10.1002/anie.201302238> and <https://doi.org/10.1039/C9CC00985J>). This impurity element is likely the factor leading to the bright spots in the Z-contrast imaging on bulk boron in **SM, Fig. S9a**.

Comment 5. The authors state that LPE of boron was sometimes found to yield ‘sheet-like’ layered products with irregular shapes. I would like to know whether the ‘sheet-like’ layered products can have impact on the Raman results for the exfoliated 2D boron. In other words, can the author provide Raman spectra of the ‘sheet-like’ layered products?

Reply 5: In the revised manuscript, we now show Raman spectra of the ‘sheet-like’ layered products in the SM, Fig. S9. As observed, the ‘sheets’ do not show the same Raman peaks as that of the ‘true’ flakes in Figure 2g, and instead display an enhanced peak matching that of the solvent used.

Comment 6. As we all know, the choice of the exfoliating solvents is crucial for the generation of nanosheets. Why did the author only choose to use acetone as the exfoliating solvent? For sample A, can nanosheets be produced in other ultrasonic solvents? If so, the conclusion of this manuscript seems to be less rigorous. Therefore, I hope that the author can supplement the experimental results of sample A-C in other types of ultrasonic solvents.

Reply 6: Past reports have shown that acetone can result in the exfoliation of sheets (<https://doi.org/10.1039/D0RA03492D> and <https://doi.org/10.1002/adma.201900353>), and being readily available, this solvent was adopted. In literature, numerous past reports have indeed studied the effects of different solvents on the liquid-phase exfoliation of boron (<https://doi.org/10.1039/C9CC00985J>, <https://doi.org/10.1002/adma.201900353>, <https://doi.org/10.1002/anie.202010723>). We chose the solvent that is proven to work, i.e. acetone, and the exploration of the effect of solvent on exfoliation efficiency is out of the scope of this.

Also, as pointed out by Reviewer 1, the well accepted reference from Backes *et al* suggests that while the yield may depend on the solvent, the size-thickness relationship (hence the mechanism) is not as dependent. We would like to reiterate that our manuscript focuses on understanding the mechanism, rather yield optimization.

In Page 6, Lines 13-14, we briefly added: “similar to previous reports which have demonstrated boron exfoliation^{19, 24}.”

Comment 7. In Introduction, the authors state that “boron nanosheets have drawn particular interest due to their potential for a range of applications, including sensors, photoelectronics, energy storage, drug delivery, bioimaging, and catalysts”. The authors should cite some related experimental breakthroughs on the application of 2D boron. For example: Nano Energy, 2022, 97, 107189; Nano Research Energy 2023, 2, e9120051; Nano Res. 2021, 14, 2337-2344; J. Mater. Chem. A 2021, 9, 13100-13108; Nano Res. 2022, 15, 2537-2544; Angew. Chem. Int. Ed. 2020, 132, 10911-10917; ACS Applied Materials & Interfaces, 2023, 15(11), 14566–14574; Nano Research, 2023,16,5826-5833.

Reply 7: Following the reviewer’s suggestion, we added the following references to Page 3, Line 10: <https://doi.org/10.1007/s12274-021-3926-6> and <https://doi.org/10.1016/j.nanoen.2022.107189>, as well as Page 3, Lines 14 and 18: <https://doi.org/10.1021/acsami.2c23234>.

Comment 8. Please ensure that the order of references corresponds to the description in the manuscript. The format of the reference needs to be in accordance with the format required by Nature Communications. There are a lot of format errors in the manuscript, for example: references 2, 7, 8 and 14.

Reply 8: We thank the reviewer for pointing out the formatting errors. The referencing in the document has been corrected.

REVIEWERS' COMMENTS

Reviewer #1 (Remarks to the Author):

The authors have thoroughly addressed all the inquiries I raised and have conducted additional measurements, which have been incorporated into the revised manuscript. Therefore, I recommend accepting this publication.

Reviewer #2 (Remarks to the Author):

The additional data strongly supports the original conclusions of the manuscript. I am pleased to see confirmation of closely spaced stacking faults capable of producing the characterized nanosheets in figure S16, and the additional low mag microscopy and XRD also supports the postulated exfoliation mechanism. Changes made in response to other reviewers concerns have also improved the manuscript. I have no remaining concerns and recommend publication.

Reviewer #3 (Remarks to the Author):

I would like to recommend publication of this manuscript in Nature Communications because all the comments have been carefully addressed.